# The long lives of primates and the 'invariant rate of ageing' hypothesis

Fernando Colchero [1,2,46 ✉], José Manuel Aburto [2,3,4], Elizabeth A. Archie[5,6], Christophe Boesch [7,8], Thomas Breuer[9,10], Fernando A. Campos [11], Anthony Collins[12], Dalia A. Conde [2,13,14], Marina Cords [15,16], Catherine Crockford[7,8], Melissa Emery Thompson [17,18], Linda M. Fedigan[19], Claudia Fichtel [20], Milou Groenenberg[9,21], Catherine Hobaiter[22,23], Peter M. Kappeler [20,24], Richard R. Lawler[25], Rebecca J. Lewis[26,27], Zarin P. Machanda[18,28], Marie L. Manguette[7,9], Martin N. Muller[17,18], Craig Packer [29], Richard J. Parnell[9], Susan Perry[30], Anne E. Pusey [31], Martha M. Robbins[7], Robert M. Seyfarth[32], Joan B. Silk[33], Johanna Staerk [2,13,14], Tara S. Stoinski[34], Emma J. Stokes[35], Karen B. Strier [36], Shirley C. Strum[37,38,39,40], Jenny Tung [31,41,42], Francisco Villavicencio [43], Roman M. Wittig [7,8], Richard W. Wrangham[18,44], Klaus Zuberbühler[22,23,45], James W. Vaupel[2,42] & Susan C. Alberts [31,41,42,46 ✉]

Is it possible to slow the rate of ageing, or do biological constraints limit its plasticity? We test the 'invariant rate of ageing' hypothesis, which posits that the rate of ageing is relatively fixed within species, with a collection of 39 human and nonhuman primate datasets across seven genera. We first recapitulate, in nonhuman primates, the highly regular relationship between life expectancy and lifespan equality seen in humans. We next demonstrate that variation in the rate of ageing within genera is orders of magnitude smaller than variation in pre-adult and age-independent mortality. Finally, we demonstrate that changes in the rate of ageing, but not other mortality parameters, produce striking, species-atypical changes in mortality patterns. Our results support the invariant rate of ageing hypothesis, implying biological constraints on how much the human rate of ageing can be slowed.

---

A full list of author affiliations appears at the end of the paper.

The maximum human life expectancy has increased since the mid-1800s by ~3 months per year[1]. These gains have resulted from shifting the majority of deaths from early to later and later ages, with no evidence of slowing the rate at which mortality increases with age (i.e. the 'rate of ageing')[2–4]. Further substantial extensions of human longevity will depend on whether it is possible to slow the rate of ageing or otherwise reduce late life mortality. Consequently, the nature of biological constraints on ageing is a central problem in the health sciences and, because of its implications for demographic patterns, is also of long-standing interest in ecology and evolutionary biology.

Across species, rates of ageing are strongly correlated with other aspects of the life history—pre-adult mortality, age at first reproduction, birth rate, metabolic rate and generation time—as well as with morphological traits such as body size and growth rate[5,6]. These correlations suggest that ageing evolves in concert with a suite of other traits, which may produce constraints on the rate of ageing within species. Indeed, researchers have long hypothesised that the rate of ageing is relatively fixed within species, not only in humans but also other animals[7–9].

This 'invariant rate of ageing' hypothesis has received mixed support. Several studies have documented a strong phylogenetic signal in the rate of ageing across multiple species of birds and mammals, suggesting biological constraints and little within-species variance in this rate[9,10]. Furthermore, Bronikowski and colleagues[11] observed greater variation in initial adult mortality than in the rate of ageing across several populations of baboons. On the other hand, across multiple mammal species, measurable differences in the rate of ageing have been documented between populations in different environments (e.g. zoo versus wild[12]).

Understanding the nature and extent of biological constraints on the rate of ageing and other aspects of age-specific mortality patterns is critical for identifying possible targets of intervention to extend human lifespans, and for understanding the evolutionary forces that have shaped lifespans within and across species. Although no consensus has been reached about the invariant rate of ageing hypothesis, further evidence that biological constraints may shape human ageing comes from the remarkably consistent relationship between life expectancy at birth ($e_0$) and lifespan equality ($\varepsilon_0$) in a diverse set of human populations[3,13,14]. While life expectancy at birth (a measure of the 'pace' of mortality[15]) describes the average lifespan in a population, lifespan equality (a measure of the 'shape' of mortality[15]) describes the spread in the distribution of ages at death in a population (see also[16,17]).

Lifespan equality is highly correlated with other measures of the distribution of ages at death, such as the coefficient of variation and the Gini coefficient, often used to measure economic inequality[13]. The distribution of ages at death tells us whether deaths are evenly distributed across the range of observed lifespans, or are concentrated around certain ages. For instance, if deaths are evenly distributed across age classes or show multiple modes, the result is high lifespan variation and low lifespan equality, while if deaths are concentrated at the tail-end of the lifespan distribution (as in most developed nations), the result is low lifespan variance and high lifespan equality. The tight positive linear relationship between life expectancy ($e_0$) and lifespan equality ($\varepsilon_0$) across a large range of human populations indicates strong but poorly understood constraints underlying variation in human mortality[3,13].

Understanding the biological constraints on ageing requires mortality data for multiple populations of nonhuman species, as well as for humans. However, comparative data across multiple populations of nonhuman animals are rarely available, making it difficult to unveil the forces underlying mortality differences within versus between species. The challenge is particularly acute for long-lived species, including nonhuman primates, the closest relatives of humans. Nonetheless, these are precisely the species that will shed most light on how biological constraints have shaped the evolution of ageing within the lineage leading to humans.

To better understand biological constraints on ageing, here we answer two questions. First, is the highly regular linear relationship between life expectancy and lifespan equality in humans also evident in other primates? Second, if so, do biological constraints on ageing underlie this highly regular relationship? To address these questions, we have assembled a large dataset on age-specific mortality rates in multiple populations of several different primate genera. Our combined dataset includes data from both wild and captive primate populations. The data from wild populations consists of individual-based birth and death data on males and females from 17 continuous long-term studies of wild primate populations representing 6 genera distributed across the order Primates, and include African monkeys (2 genera), Central and South American monkeys (1 genus), great apes (2 genera), and an indriid (1 genus, endemic to Madagascar) (Supplementary Data 1).

For these genera, we also obtained individual-based birth and death data from 13 species in zoos from Species360's Zoological Information Management System (ZIMS)[18] (see Methods, Supplementary Data 1). We also include data on a 7th primate genus, *Homo*, using male and female human mortality data from nine of the human datasets studied by Colchero et al.[13]. These nine populations had not benefited from modern advances in public health, medicine, and standards of living, enabling us to carry out the most salient comparisons with nonhuman primates. We use life tables from the Human Mortality Database[19] for **(1)** Sweden from 1751 to 1759, **(2)** Sweden in 1773, **(3)** Sweden from 1850 to 1859, **(4)** and Iceland in 1882. We also use human life tables for **(5)** England from 1600 to 1725[20], **(6)** Trinidad from 1813 to 1815[21], **(7)** Ukraine in 1933[22] and two hunter gatherer populations, **(8)** the Hadza, based on data collected between 1985 and 2000[23] and **(9)** the Ache during the pre-contact period of 1900–1978[24]. In the aggregate, our 39 combined datasets (17 wild and 13 zoo nonhuman primates, and 9 human populations; Supplementary Data 1) comprise a taxonomically diverse sample of primates and represent considerable environmental variability within genera, maximising the probability of detecting variation in ageing.

To understand potential constraints on primate ageing, we compare age-specific changes in the risk of death across multiple populations of each genus. The age-specific risk of death, often described by a hazard rate, is the basic building block of the distribution of ages at death, and therefore determines both life expectancy and lifespan equality for a population (see 'Methods'). Among most mammal species, the risk of death is high in infancy, rapidly declines during the immature period, remains relatively low until early adulthood and then rises with age as a result of senescence. This pattern can be described mathematically by the five-parameter Siler mortality function[25], given by

$$\mu(x) = \exp(a_0 - a_1 x) + c + \exp(b_0 + b_1 x), \; for \; x \geq 0 \quad (1)$$

where $a_0$, $a_1$, $c$, $b_0$, $b_1$ are mortality parameters, each of which governs different stages of the age-specific mortality. In short, parameters $a_0$ and $a_1$ drive infant and juvenile mortality, $c$ is commonly described as the age-independent mortality, and $b_0$ and $b_1$ control senescent mortality. Parameters $a_0$, $c$ and $b_0$ are scale parameters, while $a_1$ determines the speed of decline in infant and juvenile mortality and $b_1$ determines the rate of increase in adult and senescent mortality, analogous to the rate of senescence or rate of ageing.

Here, we fit these Siler models of age-specific mortality for males and females for each of the 30 non-human primate populations (Methods, Supplementary Data 1 and 2), and we examine how each of the five Siler parameters varied within and between the genera (Supplementary Figs. 1 and 2). We also calculate sex-specific values for life expectancy at birth ($e_0$) and lifespan equality ($\varepsilon_0$) in each population, and use these values to examine the relationship between life expectancy and lifespan equality within each genus (Supplementary Data 3). We conduct genus-level rather than species-level analyses because restricting ourselves to the species level severely limits the availability of individual-based datasets (e.g., among guenons, only one or two individual-based datasets are available for each species, while examining the genus provides five such datasets). We show that the highly regular linear relationship between life expectancy and lifespan equality in humans is indeed recapitulated in other primates. Further, our results suggest that the regularity in this relationship is driven by a combination, within each genus, of high variability in early and age-independent mortality and low variability in senescent mortality.

## Results

**Age-specific mortality across populations and life expectancy–lifespan equality relationship.** Our regression analyses yielded clear linear relationships between $e_0$ and $\varepsilon_0$ within each primate genus, mirroring the relationship observed within humans (Fig. 1A, B and Supplementary Fig. 3). Importantly, the linear relationship between $e_0$ and $\varepsilon_0$ is not a simple artefact of our modelling. For instance, Stroustrup et al.[26] demonstrated in laboratory experiments with *C. elegans* that changes in life expectancy occur with no change in lifespan variance. Similarly, Jones et al.[5] found no correlation between a measure of the length of life and a measure of relative variation in lifespans, across 46 species drawn from different taxa. Colchero et al.[13] found no correlation between life expectancy and lifespan equality across 15 non-primate mammal species. Aburto and van Raalte[27] showed that in Eastern European countries life expectancy and lifespan equality often moved independently of each other between the 1960s and 1980s, and van Raalte et al.[28] showed that life expectancy and lifespan equality have a negative relationship (i.e., inequality increases with life expectancy) in some human populations (Finland in the 20th and 21st centuries, in their example).

This linear relationship between life expectancy and lifespan equality emerged in our analysis despite considerable variation among populations of each genus in age-specific mortality, in the distribution of ages at death, and in the Siler mortality parameters (Supplementary Figs. 1–4, Supplementary Data 2). The slopes of these regression lines were statistically significant (i.e., $p$ value < 0.05) in five of seven genus-level datasets for females and in four of seven for males (Fig. 1A, B, Supplementary Data 4). The regression lines did not reach statistical significance in analyses that included relatively few populations or that included small or heavily censored datasets. The slopes of the regression lines were statistically significantly different than the slope of the line for humans in female sifaka, baboons, guenons and gorillas, and in male guenons, gorillas and chimpanzees.

**Drivers of the linear relationship between life expectancy and lifespan equality.** Having confirmed that the relationship between life expectancy and lifespan equality is linear and highly regular within other primate genera, as it is in humans, we next sought possible causes for this regularity. Specifically, we asked which Siler mortality parameters best explain variation among populations in life expectancy and lifespan equality, and therefore

which have a disproportionately large effect on the slopes of the regression lines. To pursue this question, we initially conducted a sensitivity analysis by simulating independent changes in each of the Siler mortality parameters (Fig. 1C) and graphically examining the effects of these changes on the life expectancy-lifespan equality relationships. Specifically, we varied one Siler parameter at a time within each genus, keeping the other four Siler parameters constant at the value found at the midpoint of the regression line.

This approach produced striking results: within each genus, simulated variation in pre-adult mortality (captured by Siler parameters $a_0$ and $a_1$) and in age-independent mortality (Siler parameter $c$) all produced lines of similar direction to the observed regression lines (Fig. 1D). That is, within the observed range of $e_0$ values, changes in these three Siler parameters resulted in $\varepsilon_0$ similar to the observed range. Therefore, consistent with theory and with the long-understood effect of averting early deaths, observed variation in life expectancy and lifespan equality within each primate genus appears to be largely accounted for by variation in the pattern of early deaths, and very little by actuarial senescence.

In stark contrast, simulated variation in the rate-of-ageing parameter (Siler parameter $b_1$) produced lines with conspicuously different direction from the observed regression lines. Specifically, changing $b_1$ moved the life expectancy–lifespan equality values away from the regression lines (Fig. 1D).

**Sensitivity of life expectancy and lifespan equality to mortality parameters.** These findings led us to postulate that, while variation in early deaths is the primary cause of observed variation in life expectancy and lifespan equality within each genus, changes in the rate of ageing in one or more populations in a genus could shift those populations towards the lines of other genera. To further investigate this possibility, we derived mathematical functions for the sensitivity of life expectancy and lifespan equality to changes in any given mortality parameter (see 'Methods'). These sensitivity functions allowed us to obtain precise measures of the amount of change in life expectancy and lifespan equality for a unit change in any given mortality parameter at any point in the life expectancy–lifespan equality landscape (including along each of the regression lines).

The resulting vectors of change (Fig. 2A) are consistent with our graphical exploration, and they also revealed the relative magnitudes of changes that each mortality parameter produces in the life expectancy–lifespan equality landscape (Fig. 2B). Specifically, a unit change in the rate of ageing parameter $b_1$ shifts the life expectancy and lifespan equality values in a direction almost perpendicular to the regression lines, and the magnitude of that change is disproportionately large compared to the other four parameters. We then calculated the degree of collinearity (how parallel versus perpendicular two vectors are) between the seven genus-specific regression lines for females and the vectors of change for each parameter. We found that the two parameters that govern infant mortality, $a_0$ and $a_1$, and the age-independent parameter $c$, produce vectors of change that are almost parallel to the regression lines. In contrast, Siler parameter $b_0$ produces vectors that are intermediate between parallel and perpendicular, while the rate-of-ageing parameter, $b_1$, produces vectors that are almost perpendicular to the regression lines (Fig. 2C). In short, changes in pre-adult mortality and in age-independent mortality tend to move a population along the regression line typical of its genus. In contrast, changes in the ageing parameters, $b_0$ and particularly $b_1$, will shift a population away from this line, into the space occupied by other genera in the landscape.

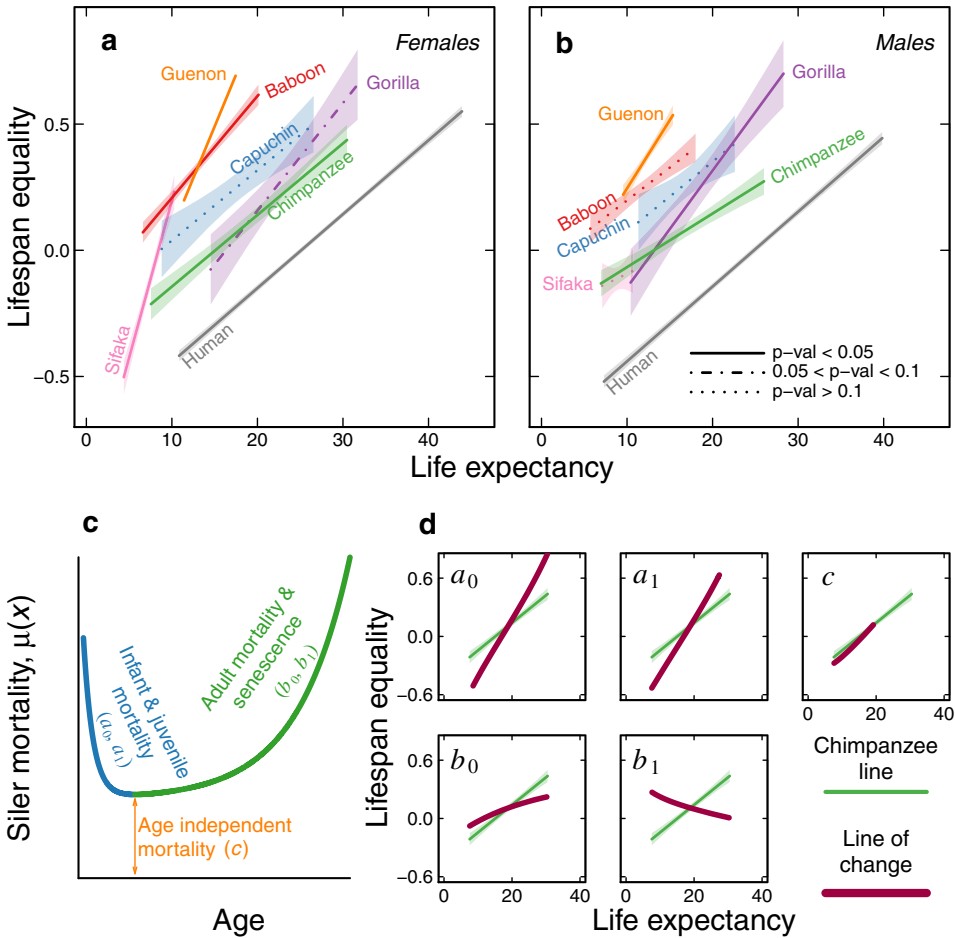

**Fig. 1 The life expectancy–lifespan equality landscape for seven genera of primates for for both sexes. a** Life expectancy and lifespan equality regression lines for females; each species is represented by a different colour. **b** Life expectancy and lifespan equality regression lines for males. Each genus is characterised by a relatively constrained relationship between life expectancy and lifespan equality, and thus a distinct regression line; colours as in **a**. The central lines are the predicted fitted values of the regression and the type of line (e.g. continuous, dashed, or dotted) depicts three levels for the p values of the slopes (how significantly different from 0 they are, two-sided t test, H$_0$: $\beta_1 = 0$, Supplementary Table 1), while the shaded polygons show the 95% confidence intervals of the regressions. **c** The relationship between the Siler mortality parameters and the resulting mortality function, given by the equation $\mu(x) = \exp(a_0 - a_1 x) + c + \exp(b_0 + b_1 x)$, where infant and juvenile mortality (blue) are controlled by parameters $a_0$ and $a_1$, age-independent mortality (orange) is captured by $c$, and senescent mortality (green) is captured by $b_0$ (initial adult mortality) and $b_1$ (rate of ageing). **d** Each box shows how gradual changes in each Siler mortality parameter modify the life expectancy and lifespan equality values (thick purple lines). The green line in each box corresponds to the regression line for female chimpanzees, shown for reference to illustrate the general trends among all genus lines. The purple curves show the changes in life expectancy and lifespan equality after varying individual Siler parameters while holding the other parameters constant. Note the striking change in life expectancy and lifespan equality that would result from changes in the ageing parameters, particularly $b_1$. See Supplementary Fig. S3 for plots that include individual points for each population. Source data to generate the regression lines are available in Supplementary Data 3.

**Amount of change in each mortality parameter along the genus lines**. If variation in pre-adult and age-independent mortality parameters account for most of the within-genus differences in life expectancy and lifespan equality, we expect the parameters that control infant and age-independent mortality to be much more highly sensitive to perturbations of $e_0$ and $\varepsilon_0$ than the parameters that control adult and senescent mortality, particularly $b_1$. To test these expectations, we quantified the relative change in each parameter along each genus line by calculating the partial derivatives of the log-transformed parameter with respect to changes in $e_0$ and $\varepsilon_0$. (see 'Methods'). These partial derivatives of the log-transformed parameter values represent standardised measures that allow direct comparison among parameters that differed in the absolute magnitude of change. We then calculated path integrals of these sensitivities along each genus line in order

to quantify the total amount of change in each parameter for all seven genera. We found that, in agreement with our previous results, in all cases the parameters that govern infant and age-independent mortality changed orders of magnitude more than those that drive adult and senescent mortality (Fig. 3).

**Discussion**

To our knowledge, our results provide the most comprehensive support to date for the idea that observed variation in mortality patterns among populations of a given genus is driven largely by changes in pre-adult mortality: previous support for this idea comes from studies of just one or a few species, typically including humans or primarily captive animal populations[3,7,10,12]. Notably, recent research on human populations[3] shows that increases in life

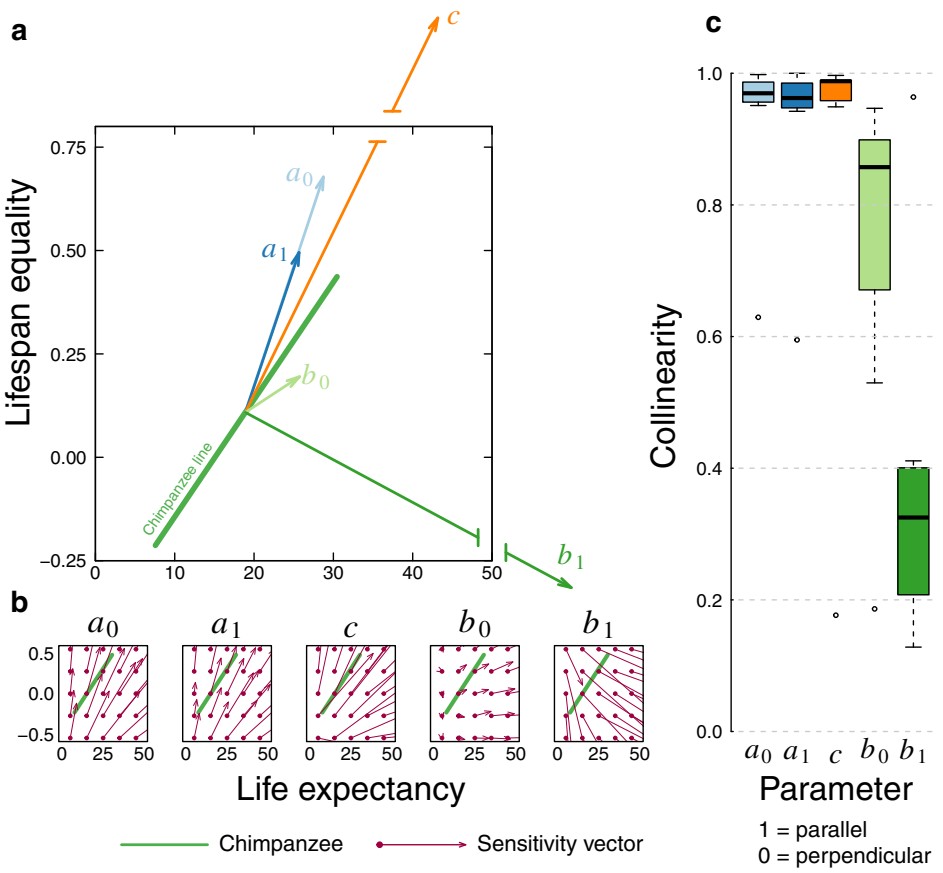

**Fig. 2 Sensitivities of life expectancy and lifespan equality to changes in mortality parameters. a** Using the female chimpanzee line (bright green) as an example, vectors depict the sensitivity at the mid-point of the genus line. Each vector depicts the direction and magnitude of change in life expectancy and lifespan equality for a unit change in the corresponding Siler mortality parameter. The x- and y-axes show the life expectancy and lifespan equality values of the sensitivity vectors for $a_0$ (light blue), $a_1$ (dark blue), and $b_0$ (light green); vectors for $c$ (orange) and $b_1$ (dark green) are particularly large, represented by broken lines (Source data are provided as a Source Data File and available in Supplementary Table 2). **b** Gradient field of sensitivities of life expectancy and lifespan equality to changes in each mortality parameter, showing the direction of change any population would experience for a given change in the parameter, from any starting point in the landscape. The green chimpanzee line is provided for reference. Each sensitivity vector (bright purple) can be interpreted as those in A, but calculated from different points on the landscape. **c** Boxplots representing the values of the seven collinearity values (one for each genus) for each of the Siler parameters for $n = 7$ independent genera. Collinearity is calculated between the mid-point of the genus line and the sensitivity vector for each parameter; a value of 1 would imply that the vector is parallel, a value of 0 would imply that it is perpendicular. Note the relatively large collinearity values for $a_0$ (light blue), $a_1$ (dark blue), and $c$ (orange), the intermediate value for $b_0$ (light green) and the relatively small value for $b_1$ (dark green). The boxplots indicate median (horizontal black line), 25th and 75th percentiles (box), the whiskers are extend to 1.5 the interquartile range, and the open points are extreme values (Source data are provided as a Source Data File and available in Supplementary Table 3).

expectancy can occur not just through decreases in pre-adult mortality but also through decreases in adult mortality. In the context of the Siler model, this would most likely translate into reductions in the $b_0$ parameter. This possibility is supported by our result that the vectors of change for Siler parameter $b_0$ produced by our sensitivity analysis are markedly less colinear with our genus-specific regression lines than the vectors of change for the pre-adult mortality parameters (Fig. 2C).

More strikingly, our results provide fresh insight into the 'invariant rate of ageing' hypothesis. In support of that hypothesis, we find that, within primate genera, rates of ageing (captured by the Siler parameter $b_1$) do indeed vary across populations, but along each genus line they generally vary orders of magnitude less than other mortality parameters (an exception is sifaka, Fig. 3). Further, our results illustrate that, within any given genus, large changes in the rate of ageing would shift a population across the life expectancy-lifespan equality landscape to a position closer to other genera. This result supports the 'invariant rate of ageing' hypothesis, although it does not rule out heterogeneity among

individuals within a population in rate of ageing. More importantly, it implicates changes in the rate of ageing as a likely source of variation in lifespan between distantly related taxa[8].

Furthermore, by considering populations exposed to a wide range of environmental conditions—from high predation and low resource availability, to unconstrained resources and veterinary care in zoos—our results have implications both for life history theory and for conservation. Life history theory predicts that among species with slow life histories (i.e., long lifespans, small litters and delayed maturity), adult survival should be buffered from environmental variability, while juvenile survival is expected to vary widely in response to the environment[29–32]. Our findings support this buffering hypothesis, in that the most dramatic observed changes in life expectancy occur because of changes in juvenile survival, while changes in adult or senescent survival account for relatively little of the observed variation within each genus.

Importantly, sufficient demographic information to understand and predict population dynamics exists for less than 1.5%

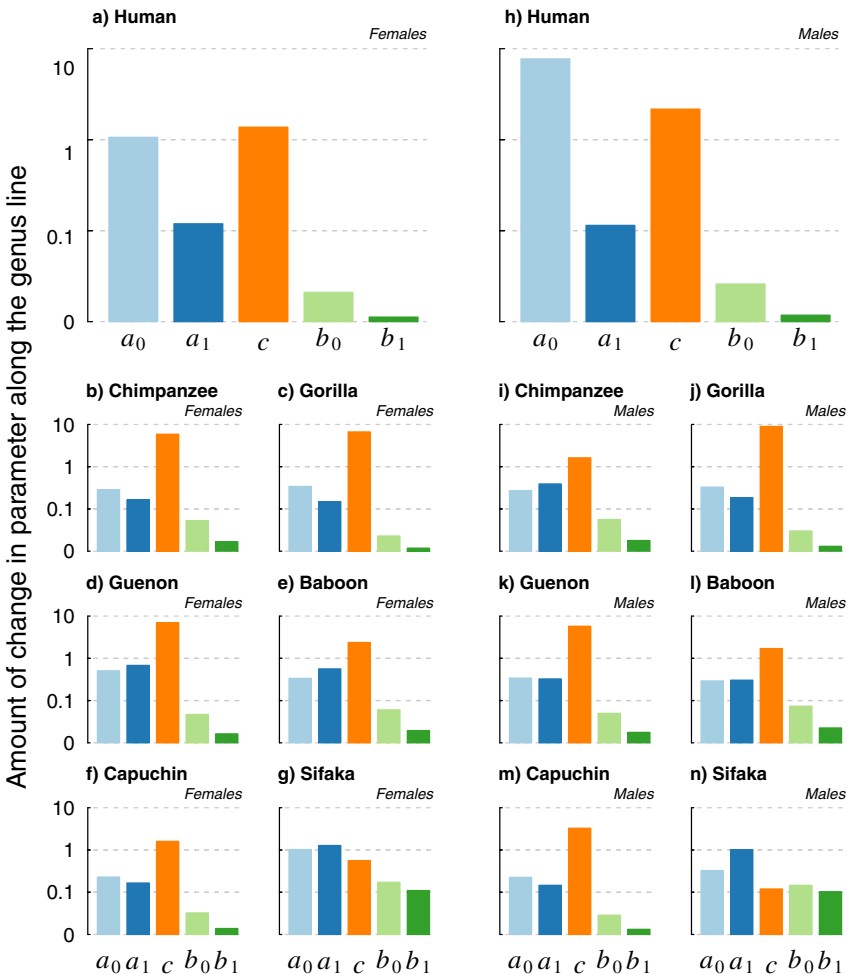

**Fig. 3 Relative magnitude of change of each parameter along the genus lines.** Pre-adult and age-independent mortality parameters ($a_0$ $a_1$, and $c$) vary several orders of magnitude more, within each genus, than the ageing parameters ($b_0$ and $b_1$). Colours: $a_0$ (light blue) $a_1$, (dark blue), $c$ (orange), $b_0$ (light green) and $b_1$ (dark green). Values were calculated by numerically solving the path integral in Eq. (9) (see 'Material and Methods') for each parameter along each genus line. The y-axes were scaled by the logarithm base 10 to improve interpretability. **a–g** depict results for females, and **h–n** for males (Source data are provided as a Source Data File and available in Supplementary Table 4).

of extant vertebrate species[33]. By unravelling the interdependence of mortality parameters within a species or genus, we can contribute to filling these glaring demographic knowledge gaps and further our understanding of the ecology and evolution of a wide range of animal species, as well as the conservation of species worldwide.

Finally, can we humans slow our own rate of ageing? Our findings support the idea that, in historical population when life expectancies were low, mortality improvements for infants, and in age-independent mortality, were the central contributors to the decades-long trend towards longer human life expectancies and greater lifespan equality[3]. These improvements were largely the result of environmental influences including social, economic, and public health advances[13,34,35]. Since the middle of the 20th century, however, declines in the baseline level of adult mortality—measured in the context of the Siler model by $b_0$— have very likely played an increasingly important role in industrialised societies[3,8]. As we show here, improvements in the environment are unlikely to translate into a substantial reduction in the rate of ageing, $b_1$, or in the dramatic increase in lifespan that would result from such a change. It remains to be seen if future advances in medicine can overcome the biological constraints that we have identified here, and achieve what evolution has not.

## Methods

**Data for non-human primates**. We obtained 30 datasets for six genera of non-human primates: sifaka (*Propithecus spp*), gracile capuchin monkey (*Cebus spp*), guenon (*Cercopithecus spp*), baboon (*Papio spp*), gorilla (*Gorilla spp*), and chimpanzee (*Pan troglodytes*) (Supplementary Data 1). Of these, 17 datasets correspond to long-term projects in the wild, while 13 were contributed by the non-profit Species360 from ZIMS[18], which is the most extensive database of life history information for animals under human care.

**Basic demographic functions**. Let $X$ be a random variable for ages at death, with observations $x \geq 0$, and let $\mu(x|\theta)$ be a continuous, non-negative parametric hazards rate or mortality function defined as

$$\mu(x, |, \theta) = \lim_{\Delta x \to 0} \frac{\Pr(x < X \leq x + \Delta x | X > x)}{\Delta x}, \qquad (2)$$

given that the limit exists, where $\theta \in \mathbb{R}^p$ is a $p$-dimensional vector of mortality parameters. The cumulative hazards rate is

$$U(x|\theta) = \int_0^x \mu(t|\theta)dt, \qquad (3)$$

which results in the survival function

$$S(x|\theta) = \exp[-U(x|\theta)]. \qquad (4)$$

The Cumulative distribution function (CDF) of ages at death is $F(x | \theta) = 1 - S(x | \theta)$, and the probability density function (PDF) of ages at death is $f(x | \theta) =$

$\mu(x \mid \boldsymbol{\theta}) S(x \mid \boldsymbol{\theta})$, for $x \geq 0$. The remaining life expectancy after age $x$ is calculated as

$$
\begin{aligned}
e(x|\boldsymbol{\theta}) &= \frac{\int_x^\infty t f(t|\boldsymbol{\theta})dt}{F(\infty) - F(x)} \\
&= \frac{\int_x^\infty S(t|\boldsymbol{\theta})dt}{S(x)},
\end{aligned}
\tag{5}
$$

which yields a life expectancy at birth given by

$$
e(0|\boldsymbol{\theta}) = \int_0^\infty S(x|\boldsymbol{\theta})dx.
\tag{6}
$$

The lifespan inequality at birth, as proposed by Demetrius[16,36] and later by Keyfitz[17], is given by

$$
\begin{aligned}
H(0|\boldsymbol{\theta}) &= -\frac{\int_0^\infty S(x|\boldsymbol{\theta})\log[S(x|\boldsymbol{\theta})]dx}{e(0|\boldsymbol{\theta})} \\
&= \frac{\int_0^\infty S(x|\boldsymbol{\theta})U(x|\boldsymbol{\theta})dx}{e(0|\boldsymbol{\theta})}.
\end{aligned}
\tag{7}
$$

Following Colchero et al.[13], we define the lifespan equality as

$$
\varepsilon(x|\boldsymbol{\theta}) = -\log[H(x|\boldsymbol{\theta})].
\tag{8}
$$

For simplicity, henceforth we note the life expectancy, lifespan inequality and lifespan equality at birth as $e(0 \mid \boldsymbol{\theta}) = e$, $H(0 \mid \boldsymbol{\theta}) = H$, and $\varepsilon(0 \mid \boldsymbol{\theta}) = \varepsilon$, respectively.

**Survival analysis.** To estimate age-specific survival for all the wild populations of non-human primates, we modified the Bayesian model developed by Colchero et al.[13] and Barthold et al.[37]. This model is particularly appropriate for primate studies that follow individuals continuously within a study area and when individuals of one or both sexes can permanently leave the study area (out-migration), while other individuals can join the study population from other areas (in-migration). Thus, it allowed us to make inferences on age-specific survival (or mortality) and on the age at out-migration.

Here we use the five parameter Siler mortality function[25], as in Eq. (1) where $\boldsymbol{\theta} = [a_0, a_0, c, b_0, b_1]$ is a vector of parameters to be estimated, and where $a_0, b_0 \in \mathbb{R}$ and $a_1, c, b_1 \geq 0$. For all species we studied, individuals of one or both sexes often leave their natal groups to join other neighbouring groups in a process commonly identified as natal dispersal. For some species, individuals who have undergone natal dispersal can then disperse additional times, described as secondary dispersal. Although dispersal within monitored groups (i.e. those belonging to the study area) does not affect the estimation of mortality, the fate of individuals that permanently leave the study area to join unmonitored groups can be mistaken for possible death. We identify this process as "out-migration", which we classify as natal or immigrant out-migration, the first for natal and the second for secondary dispersals to unmonitored groups. This distinction is particularly relevant because not all out-migrations are identified as such, and therefore the fate of some individuals is unknown after their last detection. For these individuals we define a latent out-migration state at the time they were last detected, given by the random variable indicator $O$, with observations $o_{ij} \in \{0,1\}$, where $o_{ij} = 1$ if individual $i$ out-migrated and $o_{ij} = 0$ otherwise, and where $j = 1$ denotes natal out-migration and $j = 2$ for immigrant out-migration. For known out-migrations, we automatically assign $o_{ij} = 1$. The model therefore estimates the Bernoulli probability of out-migration, $\pi_j$, such that $O_{ij} \sim \text{Bern}(\pi_j)$. Those individuals assigned as exhibiting out-migration, as well as known emigrants and immigrants, contribute to the estimation of the distribution of ages at out-migration. Here, we define a gamma-distributed random variable $V$ for ages at out-migration, with realisations $v \geq 0$, where $V_j \mid O_j = 1 \sim \text{Gam}(\gamma_{j1}, \gamma_{j2})$ and where $\gamma_{j1}, \gamma_{j2} > 0$ are parameters to be estimated with $j$ defined as above. The probability density function for the gamma distribution is $g_V(v \mid \gamma_{j1}, \gamma_{j2})$ for $v \geq 0$, with $v = x_l - \alpha_j$, where $x_l$ is the age at last detection and $\alpha_j$ is the minimum age at natal or immigrant out-migration.

In addition, since not all individuals have known birth dates, the model samples the unknown births $b_i$ as $x_{il} = t_{il} - b_i$, where $t_{il}$ is the time of last detection for individual $i$. The likelihood is then defined as

$$
p(x_{il}, x_{if}, |, \boldsymbol{\theta}, \boldsymbol{\gamma}_1, \boldsymbol{\gamma}_2, \pi_j, o_{ij}) = \begin{cases} \frac{f(x_{il})}{S(x_{if})}(1 - \pi_j) & \text{if } o_{ij} = 0 \\ \frac{S(x_{il})}{S(x_{if})}\pi_j g_V(x_{il} - \alpha_j) & \text{if } o_{ij} = 1 \end{cases},
\tag{9}
$$

where $x_{if}$ is the age at first detection, given by $x_{if} = t_{if} - b_i$, with $t_{if}$ as the corresponding time of first detection. The parameter vectors $\boldsymbol{\gamma}_1$ and $\boldsymbol{\gamma}_2$ are for natal and immigrant out-migration, respectively. In other words, individuals with $o_{ij} = 0$ are assumed to have died shortly after the last detection, while those with $o_{ij} = 1$ are censored and contribute to the estimation of the distribution of ages at out-migration. The full Bayesian posterior is then given by

$$
\begin{aligned}
p\Big(\boldsymbol{\theta}, \boldsymbol{\gamma}_1, \boldsymbol{\gamma}_2, \boldsymbol{\pi}, \mathbf{b}_u, \mathbf{o}_u, |, \mathbf{b}_k, \mathbf{o}_k, \mathbf{t}_f, \mathbf{t}_l\Big) &\propto p\Big(\mathbf{x}_l, \mathbf{x}_f, |, \boldsymbol{\theta}, \boldsymbol{\gamma}_1, \boldsymbol{\gamma}_2, \boldsymbol{\pi}, \mathbf{d}\Big) \\
&\times p(\boldsymbol{\theta})p(\boldsymbol{\gamma}_1)p(\boldsymbol{\gamma}_2)p(\boldsymbol{\pi}),
\end{aligned}
\tag{10}
$$

where the first term on the right-hand-side of Eq. (10) is the likelihood in Eq. (9), and the following terms are the priors for the unknown parameters. The vector

$\boldsymbol{\pi} = [\pi_1, \pi_2]$ is the vector of probabilities of out-migration while the subscripts $u$ and $k$ refer to unknown and known, respectively.

Following Colchero et al.[13], we used published data, expert information and an agent-based model to estimate the mortality and out-migration prior parameters for each population. We assumed a normal (or truncated normal distribution depending on the parameter's support) for all the parameters. We used vague priors for the mortality and natal out-migration parameters (sd = 10), and informative priors for the immigrant out-migration parameters (sd = 0.5). We ran six MCMC parallel chains for 25 000 iterations each with a burn-in of 5000 iterations for each population, and assessed convergence using potential scale reduction factor[38].

For the zoo data we used a simplified version of the model described above, which omitted all parts that related to out-migration. In order to produce Supplementary Figs. 1 and 2, we used the same method as for the zoo data on the human life tables. To achieve this, we created an individual level dataset from the $l_x$ column of each population, and then fitted the Siler model to this simulated data. It is important to note that the Siler model provides a close fit to the nonhuman primate data and to high-mortality human populations, although it does not provide the best fit to low-mortality human populations, in part due to the late life mortality plateau common among human populations[39] (Supplementary Fig. 6). It is therefore possible that the values of the mortality parameter $b_1$ we report in Supplementary Data 2 for the human populations are under-estimated. Nonetheless, and for the purposes of our analyses, the Siler fits to the human populations we considered here are reasonable (Supplementary Fig. 6) and we can therefore confidently state that the limitations of the Siler model do not affect the generality of our results.

**Estimation of life expectancy and lifespan equality.** Based on the results of the Bayesian inference models, we calculated life expectancy at birth as

$$
e = \int_0^\infty S(t|\hat{\boldsymbol{\theta}})dt,
\tag{11}
$$

where $S(x)$ is the cumulative survival function as defined in Eq. (4) and where $\hat{\boldsymbol{\theta}}$ is the vector of mortality parameters calculated as the mean of the conditional posterior densities from the survival analysis described above. We calculated the lifespan inequality[17,36], $H$, as

$$
H = -\frac{1}{e}\int_0^\infty S(x|\hat{\boldsymbol{\theta}})\log\Big[S(x|\hat{\boldsymbol{\theta}})\Big]dx,
\tag{12}
$$

from which we calculated lifespan equality, $\varepsilon$, as in Eq. (8). We calculated both measures for each of the study populations, and performed weighted least squares regressions for each genus, with weights given by the reciprocal of the standard error of the estimated life expectancies.

**Sensitivities of life expectancy and lifespan equality to mortality parameters.** As we mentioned above, for simplicity of notation, we will express all demographic functions by their variable notation (e.g. $e = e(0 \mid \boldsymbol{\theta})$, $S = S(x \mid \boldsymbol{\theta})$, etc.), while we will alternatively note first partial derivatives, for instance the derivative of $e$ with respect to a given mortality parameter $\theta \in \boldsymbol{\theta}$, as $e_\theta$ or $\partial e / \partial \theta$.

Proposition: If $S: \mathbb{R}_{\geq 0} \to [0, 1]$ is a continuous non-increasing parametric survival function with parameter vector $\boldsymbol{\theta}\mathbb{R}^p$, with continuous differentiable cumulative hazards function $U: \mathbb{R}_{\geq 0} \to \mathbb{R}_{\geq 0}$, and with life expetancy at birth, lifespan inequality and lifespan equality as in Eqs. (4)-(6), respectively, then the sensitivity of life expectancy, $e$, to a given parameter $\theta \in \boldsymbol{\theta}$ is

$$
e_\theta = \frac{\partial e}{\partial \theta} = \int_0^\infty S_\theta dx,
\tag{13}
$$

while the sensitivity of lifespan equality to $\theta$ is

$$
\varepsilon_\theta = \frac{\partial \varepsilon}{\partial \theta} = \frac{e_\theta(1 + H^{-1}) - H^{-1}\int_0^\infty S_\theta U dx}{e},
\tag{14}
$$

where

$$
S_\theta = \frac{\partial}{\partial \theta} S(x|\boldsymbol{\theta})
\tag{15}
$$

is the sensitivity of the survival function at age $x$ to changes in parameter $\theta$. *Proof.* The sensitivity of lifespan equality to changes in $\theta$ is derived from

$$
e_\theta = \frac{\partial}{\partial \theta}\int_0^\infty S dx,
\tag{16}
$$

which, by Leibnitz's rule, Eq. (16) becomes

$$
e_\theta = \int_0^\infty \frac{\partial S}{\partial \theta}dx = \int_0^\infty S_\theta dx.
\tag{17}
$$

The sensitivity of lifespan equality to changes in $\theta$ can be calculated as

$$\varepsilon_\theta = \frac{\partial}{\partial \theta}(-\log H)$$
$$= -\frac{\partial}{\partial \theta}\log H$$
$$= -\frac{1}{H}\frac{\partial H}{\partial \theta} \qquad (18)$$
$$= -\frac{1}{H}\frac{\partial}{\partial \theta}\left(\frac{\int_0^\infty SU dx}{e}\right).$$

By the quotient and Leibnitz's rules, Eq. (18) can be modified as

$$\varepsilon_\theta = -\frac{1}{He^2}\left[\frac{\partial}{\partial \theta}\left(\int_0^\infty SU dx\right)e - \left(\int_0^\infty SU dx\right)\frac{\partial e}{\partial \theta}\right]$$
$$= -\frac{1}{He}\int_0^\infty \frac{\partial}{\partial \theta}(SU)dx + \frac{1}{He}\frac{\int_0^\infty SU dx}{e}\frac{\partial e}{\partial \theta}. \qquad (19)$$

The first term in Eq. (19) can be further decomposed by the product rule, while the second term can be modified following the equality for $H$ in Eq. (7), which yields

$$\varepsilon_\theta = -\frac{1}{He}\int_0^\infty \left(\frac{\partial S}{\partial \theta}U + S\frac{\partial U}{\partial \theta}\right)dx + \frac{1}{e}e_\theta$$
$$= -\frac{1}{He}\left(\int_0^\infty S_\theta U dx + \int_0^\infty S\frac{\partial U}{\partial \theta}dx\right) + \frac{1}{e}e_\theta. \qquad (20)$$

By the chain rule, we have that $\frac{\partial U}{\partial \theta} = -\frac{\partial}{\partial \theta}\log S = -\frac{1}{S}\frac{\partial S}{\partial \theta}$, which modifies Eq. (20) as

$$\varepsilon_\theta = -\frac{1}{He}\left(\int_0^\infty S_\theta U dx - \int_0^\infty \frac{\partial S}{\partial \theta}dx\right) + \frac{1}{e}e_\theta$$
$$= -\frac{1}{He}\left(\int_0^\infty S_\theta U dx - e_\theta\right) + \frac{1}{e}e_\theta$$
$$= -\frac{\int_0^\infty S_\theta U dx}{He} + \frac{e_\theta}{e}\left(1 + \frac{1}{H}\right) \qquad (21)$$
$$= \frac{e_\theta\left(1 + H^{-1}\right) - H^{-1}\int_0^\infty S_\theta U dx}{e},$$

hence completing the proof. ∎

**Changes in parameters along the genus lines**. From the results in Eqs. (13) and (14), we calculated the vectors of change (gradient vectors) at any point $\langle e_j, \varepsilon_j\rangle$ of the life expectancy-lifespan equality landscape, as a function of each of the Siler mortality parameters (See Fig. 2A, B).

To quantify the amount of change of each parameter along the genus lines, we derived the sensitivities of a given mortality parameter $\theta$ to changes in life expectancy and lifespan equality, namely $\frac{\partial \theta}{\partial e} = \frac{1}{e_\theta}$ for $e_\theta \neq 0$, and $\frac{\partial \theta}{\partial \varepsilon} = \frac{1}{\varepsilon_\theta}$ for $\varepsilon_\theta \neq 0$. With these sensitivities we calculated the gradient vector

$$\nabla \theta = \left\langle \frac{\partial \theta}{\partial e}, \frac{\partial \theta}{\partial \varepsilon}\right\rangle \qquad (22)$$

for any parameter at any point along the genus lines. Here we find a linear relationship between life expectancy and lifespan equality, given by

$$m(e_{ik}) = \hat{\varepsilon}_{ik} = \beta_{0k} + \beta_{1k}e_{ik}, \qquad (23)$$

for $i = 1, \ldots, n_k$, where $n_k$ is the number of populations for genus $k$, and $\hat{\varepsilon}_{ik}$ is the fitted value of lifespan equality for population $i$ in genus $k$, and $\beta_{0k}$ and $\beta_{1k}$ are linear regresssion parameters for genus $k$. To estimate the amount of change in parameter $\theta$ along the line for genus $k$, we can solve the path integral

$$\Theta_k = \int_{C_k} \nabla \theta d\mathbf{r}, \qquad (24)$$

where path $C_k$ is determined by the linear model for genus $k$ and $d\mathbf{r} = \langle de, d\hat{\varepsilon}\rangle = \langle de, d\,m(e)\rangle$ is the rate of change in the velocity vector $\mathbf{r} = \langle e, \hat{\varepsilon}\rangle = \langle e, m(e)\rangle$.

In order to compare results between the different mortality parameters in vector $\boldsymbol{\theta}$, we use the transformation $g(\theta) = \log \theta$, which yields the following partial derivatives

$$\frac{\partial}{\partial e}g(\theta) = \frac{1}{\theta}\frac{\partial \theta}{\partial e} \qquad (25)$$

and

$$\frac{\partial}{\partial \varepsilon}g(\theta) = \frac{1}{\theta}\frac{\partial \theta}{\partial \varepsilon}. \qquad (26)$$

Thus the gradient vector becomes

$$\nabla \theta = \left\langle \frac{\partial}{\partial e}g(\theta), \frac{\partial}{\partial \varepsilon}g(\theta)\right\rangle \qquad (27)$$

while the path integral in Eq. (24) is modified accordingly. In short, the path integral $\Theta_j$ provides a measure of the relative change in parameter $\theta$ along the genus line (Fig. 3). To allow comparisons between all genera, we scaled the values of each path integral by the length of each line.

**Applications to the Siler mortality model**. The Cumulative hazards for the Siler mortality model in Eq. (7) is given by

$$U(x) = \frac{e^{a_0}}{a_1}(1 - e^{-a_1 x}) + cx + \frac{e^{b_0}}{b_1}\left(e^{b_1 x} - 1\right), \qquad (28)$$

The sensitivities in Eqs. (13) and (14) require calculating $S_\theta$ for all $\theta \in \boldsymbol{\theta}$. Treating $S(x)$ as the function composition $W(V)$, where $W = \exp(x)$ and $V = -U$, then $S_\theta$ is

$$S_\theta = \frac{dW}{dV}V_\theta = -SU_\theta, \qquad (29)$$

where $U_\theta$ is the first derivative of $U(x \mid \boldsymbol{\theta})$ with respect to $\theta$. For each of the Siler mortality parameters, we then have

$$S_{a_0} = S(x|\boldsymbol{\theta})\frac{e^{a_0}}{a_1}(e^{-a_1 x} - 1) \qquad (30)$$

$$S_{a_1} = S(x|\boldsymbol{\theta})\frac{e^{a_0}}{a_1}\left[\frac{1}{a_1} - e^{-a_1 x}\left(x + \frac{1}{a_1}\right)\right] \qquad (31)$$

$$S_c = -S(x|\boldsymbol{\theta})x \qquad (32)$$

$$S_{b_0} = S(x|\boldsymbol{\theta})\frac{e^{b_0}}{b_1}\left(1 - e^{b_1 x}\right) \qquad (33)$$

$$S_{b_1} = S(x|\boldsymbol{\theta})\left[e^{b_1 x}\left(\frac{1}{b_1} - x\right) - \frac{1}{b_1}\right]. \qquad (34)$$

All analyses were performed in the free open source programme R[40]. The R functions we created for this project can be found in[41].

**Reporting summary**. Further information on research design is available in the Nature Research Reporting Summary linked to this article.

## Data availability

Full datasets for survival analyses on wild populations and populations under human care of non-human primate species supporting the findings of this study were used under license for the current study and are not publicly available; specific requests for the access to the wild data should be addressed to the PIs of the data. Data of animals under human care are however available from Species360 (https://www.species360.org/) upon reasonable request. Summarised data underlying the analyses here and sufficient to calculate life tables and summary statistics such as life expectancy and lifespan equality are available in the Dryad data repository, URL: https://doi.org/10.5061/dryad.4b8gthtb4. Data from human populations were obtained from the Human Mortality Database (https://www.mortality.org/) and published sources. Source data are provided with this paper.

## Code availability

The code showing an overview of the analyses used in the manuscript is available at https://github.com/fercol/ColcheroEtal2021NatComm (https://doi.org/10.5281/zenodo.4736892).

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

## Acknowledgements

The governments of Botswana, Brazil, Costa Rica, Côte d'Ivoire, Kenya, Madagascar, Uganda, Republic of Congo, Rwanda and Tanzania provided permission for the primate field studies; all research complied with guidelines in the host countries. We thank the zoo and aquarium staff for manageing their animal records in Zoological Information Management System (ZIMS) and providing high quality demographic data for this project (Data Use Approval Number 82421). Duke University, Max Planck Institute of Demographic Research, and University of Southern Denmark provided logistical support. Annette Baudisch provided valuable feedback on the manuscript. This work was supported by NIA P01AG031719 to J.W.V. and S.C.A., with additional support provided by the Max Planck Institute of Demographic Research and the Duke University Population Research Institute.

## Author contributions

F.C. contributed Conceptualisation, Methodology, Formal Analysis, Visualisation, Writing—Original and Writing-Review/editing, Project Administration. J.M.A. and F.V. contributed Methodology, Writing—Review/editing. J.W.V. contributed Conceptualisation, Writing—Review/editing, Funding Acquisition. S.C.A. contributed Conceptualisation, Methodology, Resources, Writing—Original and Writing—Review/editing, Visualisation, Project Administration, Funding Acquisition. E.A.A., C.B., T.B., F.A.C., A.C., D.A.C., M.C., C.C., M.E.T., L.M.F., C.F., M.G., C.H., P.M.K., R.R.L., R.J.L., Z.P.M., M.L.M., M.N.M., C.P., R.J.P., S.P., A.E.P., M.M.R., R.M.S., J.B.S., J.S.,T.S.S., E.J.S., K.B.S., S.C.S., J.T., R.M.W., R.W.W. and K.Z. contributed resources, Methodology, Writing—Review/editing.

## Competing interests

The authors declare no competing interests.

## Additional information

[1]Department of Mathematics and Computer Science, University of Southern Denmark, Odense, Denmark. [2]Interdisciplinary Centre on Population Dynamics, University of Southern Denmark, Odense, Denmark. [3]Department of Sociology, Leverhulme Centre for Demographic Science, Nuffield College at University of Oxford, Oxford, UK. [4]Lifespan Inequalities Research Group, Max Planck Institute for Demographic Research, Rostock, Germany. [5]Department of Biological Sciences, University of Notre Dame, Notre Dame, IN, USA. [6]Institute of Primate Research, National Museums of Kenya, Nairobi, Kenya. [7]Max Planck Institute for Evolutionary Anthropology, Leipzig, Germany. [8]Taï Chimpanzee Project, CSRS, Abidjan, Côte d'Ivoire. [9]Mbeli Bai Study, Wildlife Conservation Society Congo Program, Brazzaville, Congo. [10]World Wide Fund for Nature - Germany, Berlin, Germany. [11]Department of Anthropology, University of Texas at San Antonio, San Antonio, TX, USA. [12]Gombe Stream Research Centre, Jane Goodall Institute, Kigoma, Tanzania. [13]Species360 Conservation Science Alliance, Bloomington, MN, USA. [14]Department of Biology, University of Southern Denmark, Odense, Denmark. [15]Department of Ecology, Evolution, and Environmental Biology, Columbia University, New York, NY, USA. [16]New York Consortium in Evolutionary Anthropology, New York, NY, USA. [17]Department of Anthropology, University of New Mexico, Albuquerque, NM, USA. [18]Kibale Chimpanzee Project, Fort Portal, Uganda. [19]Department of Anthropology and Archaeology, University of Calgary, Calgary, AB, Canada. [20]Behavioral Ecology & Sociobiology Unit, German Primate Center, Leibniz Institute for Primate Research, Göttingen, Germany. [21]World Wide Fund for Nature- Cambodia Program, Phnom Penh, Cambodia. [22]School of Psychology and Neuroscience, University of St Andrews, St Andrews, Scotland, UK. [23]Budongo Conservation Field Station, Masindi, Uganda. [24]Department for Sociobiology/ Anthropology, Johann-Friedrich-Blumenbach Institute of Zoology and Anthropology, University of Göttingen, Göttingen, Germany. [25]Department of Sociology and Anthropology, James Madison University, Harrisonburg, VA, USA. [26]Department of Anthropology, University of Texas at Austin, Austin, TX, USA. [27]Ankoatsifaka Research Station, Morondava, Madagascar. [28]Departments of Anthropology and Biology, Tufts University, Medford, MA, USA. [29]College of Biological Sciences, Department of Ecology, Evolution and Behavior, University of Minnesota, Saint Paul, MN, USA. [30]Department of Anthropology, and Behavior, Evolution & Culture Program, UCLA, Los Angeles, CA, USA. [31]Department of Evolutionary Anthropology, Duke University, Durham, NC, USA. [32]Department of Psychology, University of Pennsylvania, Philadelphia, PA, USA. [33]School of Human Evolution and Social Change, Institute of Human Origins, Arizona State University, Tempe, AZ, USA. [34]Dian Fossey Gorilla Fund International, Atlanta, GA, USA. [35]Wildlife Conservation Society, Global Conservation Program, Bronx, NY, USA. [36]Department of Anthropology, University of Wisconsin-Madison, Madison, WI, USA. [37]Department of Anthropology, University of California, San Diego, La Jolla, CA, USA. [38]Uaso Ngiro Baboon Project, Laikipia, Kenya. [39]Kenya Wildlife Service, Nairobi, Kenya. [40]African Conservation Centre, Nairobi, Kenya. [41]Department of Biology, Duke University, Durham, NC, USA. [42]Duke Population Research Institute, Duke University, Durham, NC, USA. [43]Department of International Health, Bloomberg School of Public Health, Johns Hopkins University, Baltimore, MD, USA. [44]Department of Human Evolutionary Biology, Harvard University, Cambridge, MA, USA. [45]Institute of Biology, University of Neuchâtel, Neuchâtel, Switzerland. [46]These authors contributed equally: Fernando Colchero, Susan C. Alberts. ✉email: colchero@imada.sdu.dk; alberts@duke.edu

