## [Peer Review File · Nature Communications]

Reviewers' Comments:

Reviewer #1:

Remarks to the Author:

An impressive paper, interesting approach and results that are worth publication. But the results lead the authors to claims that are incorrect or at least overblown. A much more careful and precise statement of the findings and implications is essential. The other ideas are best presented as hypotheses, with some or no support.

This is an impressive effort with a large and probably unrivalled collection of human and non-human primate data. The analytic approach that unifies all this here is the Siler model, that is fit using a Bayesian approach, so that the parameters of the model can be used to compare otherwise disparate species.

The authors state that they have previously found a "remarkably consistent relationship between life expectancy at birth and lifespan equality" – I would agree that they have found empirical support but the idea is surely expected. For one thing, even the simplest model where individuals either die at birth with probability q , or live on and die at a fixed age T , shows that life expectancy is $(1-q)T$ whereas the variance in age at death is $q(1-q)T^2$. So the correlation must depend on q . For another thing, ecologists have known for decades that early survival has the largest effect on life expectancy, and similarly on the variation in age at death.

The authors highlight at least three conclusions. (1) They say "variation in mortality patterns among populations of a given genus is driven largely by changes in pre-adult mortality". I would argue that this result is expected – which is fine, in my view!

(2) They say that their results, especially their sensitivity analysis, provide strong support for the 'invariant rate of aging' hypothesis. Certainly they show that equality and life expectancy changes with b_1 are not the same as for the other parameters. But surely this is just a consequence of the Gompertz-like behavior of the Siler model at high ages (which means that life expectancy goes like $-\log(b_1)/b_1$ whereas the variance thereabouts goes as $(1/b_1)$.)

(3) They argue that "environmentally-influenced infant and age-independent mortality improvements were the central contributor to the decades-long trend towards longer human life expectancies..." "declines in the baseline level of adult mortality, b_0 , have played an increasingly important role and, "improvements in the environment are unlikely to translate into a substantial reduction in the rate of aging."

I do not see why the results here support these claims. Numerous studies have found that human mortality at old ages has declined dramatically over time (including work by at least one of these authors!). In addition, although demographers seem to have an inordinate fondness for simple models like the Siler or the Gompertz, such models do not work for human data over the last century unless we suppose that ALL parameters change.

Reviewer #2:

Remarks to the Author:

Are there biological constraints to the rate of aging? This is a central question that interests not only demographers and evolutionary biologists, but it is of fundamental significance to all readers who are interested in human aging. The best way to understand the evolution of human aging is to incorporate humans into the context of other nonhuman species, but top-quality data for other species is rarely available. This paper, by Colchero et al., uses extraordinary datasets with detailed data on individual-based birth and death from long-term wild and zoo nonhuman primates, and from human populations.

These new results show that the linear relationship between life-expectancy and lifespan-equality is consistent across primates. They also show that variation in mortality patterns is determined primarily by pre-adult and age-independent mortality and less by variation in the rates of aging. In other words, the rate of aging is relatively invariant compared to environmental effects on mortality. This comparative analysis, across a diverse set of species and environments, provides results that are a major step forward in our understanding of the plasticity of the demographic patterns of aging. The methods and analysis are very clear, and it is written in such a way that the work is accessible to readers outside the field.

The title of this paper is 'The evolutionary landscape of primate longevity', yet this paper is focused on the invariance of the rates of aging across different species. These results have clear implications for longevity, but I hope that the authors will consider revising the title to better reflect the analysis.

There is a minor typo on line 525, otherwise I have no other suggestions for revision.

REVIEWER COMMENTS

We would like to thank the editor and both reviewers for their very constructive comments. We hope that we have addressed satisfactorily your comments. Please note that the changes on the main text in response to your comments are highlighted in yellow. Please find below our detailed answers to your comments.

Reviewer #1 (Remarks to the Author):

An impressive paper, interesting approach and results that are worth publication. But the results lead the authors to claims that are incorrect or at least overblown. A much more careful and precise statement of the findings and implications is essential. The other ideas are best presented as hypotheses, with some or no support.

Response: We are grateful for your positive comments, and hope that our revisions fully address your concerns.

This is an impressive effort with a large and probably unrivalled collection of human and non-human primate data. The analytic approach that unifies all this here is the Siler model, that is fit using a Bayesian approach, so that the parameters of the model can be used to compare otherwise disparate species.

The authors state that they have previously found a “remarkably consistent relationship between life expectancy at birth and lifespan equality” – I would agree that they have found empirical support but the idea is surely expected. For one thing, even the simplest model where individuals either die at birth with probability q , or live on and die at a fixed age T , shows that life expectancy is $(1-q)T$ whereas the variance in age at death is $q(1-q) T^2$. So the correlation must depend on q . For another thing, ecologists have known for decades that early survival has the largest effect on life expectancy, and similarly on the variation in age at death.

Response: Thank you for these comments. We agree entirely with your point that it is well known that early survival has a large effect on life expectancy and the variance in ages at death. We now clarify that this result is well-established (lines 226-227), and that the novelty of our result lies partly in demonstrating the highly regular manner in which the various mortality parameters -- not only early survival but all mortality parameters -- contribute to both within *and* between-genus relationships between life expectancy and lifespan equality. We also highlight our empirical demonstration of differences between primate genera in the slopes and intercepts of the regression lines (lines 208-210).

We also now clarify, using examples from other studies, that the strong linear relationship between life expectancy and lifespan equality is not inevitable (lines 191-200) as follows: “For instance, Stroustrup *et al.*²⁶ demonstrated in laboratory experiments with *C. elegans* that changes in life expectancy occur with no change in lifespan variance. Similarly, Jones *et al.*⁵ found no correlation between a measure of the length of life and a measure of relative variation in lifespans, across 46 species drawn from different taxa. Colchero *et al.*¹³ find no correlation between life expectancy and lifespan equality across 15 non-primate mammal species. Aburto & van Raalte²⁷ show that in Eastern European countries life expectancy and lifespan equality often moved independently of each other between the 1960s and 1980s, and van Raalte *et al.*²⁸ show that life expectancy and lifespan equality have a *negative* relationship (i.e., *inequality* increases with life expectancy) in some human populations (Finland in the 20th and 21st centuries, in their example).” In other words, the linear relationship between life

expectancy and lifespan equality is neither an inevitable relationship, nor a spurious product of our model, but most likely driven by natural processes.

Finally, the specific model you propose produces a quadratic relationship between the mean and the variance as q changes. To further illustrate our point that the positive relationship we found is not inevitable, we propose an equally simple model of mortality, where the hazard rate, $\mu(x)$, is set at a constant value b , thus $\mu(x) = b$. In addition, we consider not only the variance, but also the measure of variation we used, derived from the life table inequality proposed by Demetrius^{16,38} and Keyfitz¹⁷, which is closely related to the coefficient of variation (CV) rather than the variance. In our simple model, life expectancy would be $E[X] = e_0 = 1/b$, while the variance would be $\text{Var}[X] = 1/b^2$ and the coefficient of variation would be $\text{CV} = \sqrt{\text{Var}[X]} / e_0 = 1$. Our measure of lifespan equality, ε (closely related to $-\ln(\text{CV})$), in this case would therefore simply equal 0 for all values of b and, consequently, for all values of e_0 . In short, using this simple model of mortality, it is not possible to find a linear relationship between life expectancy and lifespan equality. In contrast, in the empirical data we find a strong, positive, and highly linear relationship.

The authors highlight at least three conclusions. (1) They say “variation in mortality patterns among populations of a given genus is driven largely by changes in pre-adult mortality”. I would argue that this result is expected – which is fine, in my view!

Response: We agree!

(2) They say that their results, especially their sensitivity analysis, provide strong support for the ‘invariant rate of aging’ hypothesis. Certainly they show that equality and life expectancy changes with b_1 are not the same as for the other parameters. But surely this is just a consequence of the Gompertz-like behavior of the Siler model at high ages (which means that life expectancy goes like $-\log(b_1)/b_1$ whereas the variance thereabouts goes as $(1/b_1)$.)

Response: If we understand correctly, your concern is that b_1 naturally varies less than the other parameters, and we agree. This was the rationale behind our use of a scaled measure of change, relative to the magnitude of the parameter, rather than an absolute measure of change. That is, we analyzed the amount of change per parameter along the genus lines by taking the sensitivity of the parameters to changes in life expectancy and lifespan equality, *relative to the parameter’s magnitude* (lines 264-269). We did this precisely to avoid comparing parameters that can naturally vary more than others, as you point out. Even when we scale by relative magnitudes, our results still support the conclusion that b_1 varies less than other parameters (Figure 3). In addition, the relatively small variation in b_1 is not inevitable -- for instance in the sifaka, b_1 varies almost as much as c .

(3) They argue that “environmentally-influenced infant and age-independent mortality improvements were the central contributor to the decades-long trend towards longer human life expectancies...”

“ declines in the baseline level of adult mortality, b_0 , have played an increasingly important role

and, “ improvements in the environment are unlikely to translate into a substantial reduction in the rate of aging.”

I do not see why the results here support these claims. Numerous studies have found that human mortality at old ages has declined dramatically over time (including work by at least one of these authors!). In addition, although demographers seem to have an inordinate

fondness for simple models like the Siler or the Gompertz, such models do not work for human data over the last century unless we suppose that ALL parameters change.

Response: We agree that our data do not directly support claims about changes in life expectancy in human populations over the past century. As you mention, these declines have been documented by some of us (Vaupel, Aburto), and by other demographers. In particular, we refer to recent work by Aburto *et al.* (2020, PNAS, reference 3 in the main text), where the authors find that recent declines in old age mortality is most likely due to an overall reduction in the level of mortality -- not only in age-independent mortality, but also very likely in the second Gompertz baseline parameter, b_0 . However, we agree that we have not proven this last statement for human populations in post-industrialization societies, so we have modified the text as follows: “Since the middle of the 20th century, however, declines in the baseline level of adult mortality— measured in the context of the Siler model by b_0 —have very likely played an increasingly important role in industrialized societies ^{3,8}” (lines 316-318).

With respect to our statement that the environmental context has played a major role in reducing infant and age-independent mortality over time, our claim is based on the very wide range of environmental conditions experienced by the different populations we studied here. The nonhuman primate populations represent a spectrum from wild populations with high exposure to predation and disease, to wild populations in more benign environments, to captive populations with high-quality medical care. We point to this rationale in lines 296-299.

Finally, we understand and agree with your reservations about using the Siler model for industrialized human populations over the last century. However, as we show in Fig. S6, the Siler model fits the high mortality human populations we studied here (which do not represent industrialized, low-mortality human populations from the past century very well). We now note this in the Methods as well, lines 393-401.

Reviewer #2 (Remarks to the Author):

Are there biological constraints to the rate of aging? This is a central question that interests not only demographers and evolutionary biologists, but it is of fundamental significance to all readers who are interested in human aging. The best way to understand the evolution of human aging is to incorporate humans into the context of other nonhuman species, but top-quality data for other species is rarely available. This paper, by Colchero *et al.*, uses extraordinary datasets with detailed data on individual-based birth and death from long-term wild and zoo nonhuman primates, and from human populations. These new results show that the linear relationship between life-expectancy and lifespan-equality is consistent across primates. They also show that variation in mortality patterns is determined primarily by pre-adult and age-independent mortality and less by variation in the rates of aging. In other words, the rate of aging is relatively invariant compared to environmental effects on mortality. This comparative analysis, across a diverse set of species and environments, provides results that are a major step forward in our understanding of the plasticity of the demographic patterns of aging. The methods and analysis are very clear, and it is written in such a way that the work is accessible to readers outside the field.

Response: We are very grateful for your encouraging comments.

The title of this paper is ‘The evolutionary landscape of primate longevity’, yet this paper is focused on the invariance of the rates of aging across different species. These results have clear implications for longevity, but I hope that the authors will consider revising the title to better reflect the analysis.

Response: We agree with your feedback and have changed the title accordingly to “The long lives of primates: implications for the ‘invariant rate of aging’ hypothesis.”

There is a minor typo on line 525, otherwise I have no other suggestions for revision.

Response: Thank you for pointing out the typo, we have corrected it.

Reviewers' Comments:

Reviewer #1:

Remarks to the Author:

The authors have done a good job of responding to my critique. In particular they have provided reasons to believe that their central assertion about aging rate is unexpected and widespread.

Shripad Tuljapurkar